# LABEL-FREE PRIVACY-PRESERVING LEARNING FOR ZERO-SHOT ACTION RECOGNITION

## ABSTRACT

Traditional action recognition relies on labeled data and closed-set assumptions, limiting adaptability to novel actions and environments. Vision-Language Models (VLMs) offer a more flexible alternative through text-image alignment, enabling zero-shot action recognition. However, using raw video data poses privacy risks due to sensitive visual content. Privacy-Preserving Action Recognition (PPAR) aims to anonymize videos while preserving action-relevant semantics. Existing learning-based PPAR approaches often require both action and privacy annotations and retraining of recognition models on anonymized data, limiting their flexibility and compatibility with powerful pretrained VLMs. We propose LaF-Privacy, a novel label-free privacy-preserving framework for zero-shot action recognition. Our method is trained without any manual annotations, using two complementary objectives: preserving high-level action-relevant features and suppressing low-level appearance cues between raw and anonymized videos. We adopt a video transformer encoder for spatio-temporal learning and introduce an Action-Aware Masking Module (AAMM) to discard irrelevant regions, further enhancing privacy. LaF-Privacy enables direct use of pretrained VLMs for zero-shot inference on anonymized videos. Experiments on VP-UCF101 and VP-HMDB51 demonstrate that our approach achieves state-of-the-art trade-offs between privacy protection and zero-shot recognition performance.

## 1 INTRODUCTION

Action recognition is a long-standing task in computer vision. Traditional methods (Carreira & Zisserman, 2017; Tran et al., 2018) classify videos into a fixed set of categories and perform well in fully supervised settings, but they lack flexibility and generalization to unseen actions or environments, often requiring costly retraining with labeled data. Vision-Language Models (VLMs) (Radford et al., 2021; Jia et al., 2021) offer a more generalizable alternative by aligning visual inputs with natural language through contrastive learning on large-scale image-text pairs. Subsequent works (Wang et al., 2021; Ni et al., 2022) extend VLMs to video action recognition, enabling strong performance and zero-shot recognition, where actions can be identified based solely on textual prompts without any task-specific fine-tuning. This makes VLMs well-suited for open-world settings with dynamic categories and limited annotations.

Action recognition has a wide range of applications, from surveillance and healthcare to human-computer interaction. However, as public awareness of privacy issues continues to grow, the use of raw video data for action recognition raises serious concerns. Videos often contain sensitive visual information such as skin color, facial features, gender, nudity, and interpersonal relationships (Wu et al., 2020), all of which may lead to privacy leakage. This has motivated a new research direction, **Privacy-Preserving Action Recognition (PPAR)**. In this task, input videos are anonymized through various transformation techniques with two primary objectives. The first is action recognition, ensuring the transformed videos preserve sufficient semantic cues for accurate recognition. The second is privacy protection, effectively removing identifiable appearance-related information so that sensitive attributes cannot be perceived by either humans or machines. These two goals often conflict with each other. Enhancing privacy may degrade recognition performance and vice versa, making it crucial to find a meaningful trade-off between utility and privacy in PPAR systems.

Previous PPAR approaches can be broadly categorized into three types. Downsampling-based methods (Butler et al., 2015; Ryoo et al., 2017; 2018; Srivastav et al., 2019) reduce resolution or apply blurring to obscure sensitive information. Obfuscation-based methods (Ren et al., 2018; Zhang et al., 2021; Ilic et al., 2024) detect and mask regions that may reveal private attributes, such as human faces. While both reduce identifiable details, they often impair the video's action-relevant features, making it difficult to maintain a good balance between recognition performance and privacy protection. Recent research has therefore turned to learning-based approaches (Wu et al., 2020; 2018; Kumawat & Nagahara, 2022; Dave et al., 2022; Li et al., 2023; Peng et al., 2023; Li et al., 2024), which jointly train anonymization modules with action and privacy recognition models. During training, anonymized videos are fed to both models, and the network is optimized through adversarial learning by minimizing the action loss while maximizing the privacy loss. This strategy enables the model to selectively preserve motion cues essential for action understanding while actively suppressing appearance-based identity features, leading to a more effective trade-off between utility and privacy.

However, existing learning-based PPAR frameworks still face several critical limitations, as shown in Figure 1 (a). First, they typically require both action and privacy annotations for every training video, which can be impractical in real-world applications. While action labels are already costly to obtain, collecting privacy-related annotations demands even greater effort and domain expertise, significantly increasing the annotation burden. Second, after anonymization, these approaches generally require retraining the action recognition model on the anonymized videos in order to maintain classification performance. Since the anonymized content differs significantly from the original visual input, models trained on original videos cannot be directly applied. This prevents the use of powerful pretrained models, such as Vision-Language Models (VLMs), and makes zero-shot action recognition infeasible. Furthermore, when new action categories or novel scenes emerge, the entire pipeline must be retrained, greatly limiting scalability and flexibility in real-world settings.

To address the limitations of prior approaches, we propose LaF-Privacy, a label-free privacy-preserving framework that eliminates the need for manual annotations and supports zero-shot inference, as illustrated in Figure 1 (b). Instead of relying on action or privacy labels, our method is trained in an unsupervised setting using two complementary objectives. First, we encourage the anonymized video to **preserve high-level action-relevant features** by minimizing the distance between deep representations of the raw and anonymized videos. This ensures that essential semantics for action recognition are retained. Second, we enforce **visual dissimilarity in low-level appearance features** between the raw and anonymized videos, promoting effective privacy protection. LaF-Privacy adopts a video transformer encoder for learning spatiotemporal representations and anonymizing the entire video sequence. Additionally, we introduce an Action-Aware Masking Module (AAMM) that removes regions irrelevant to action understanding, further enhancing privacy protection without degrading utility. Our framework offers two major advantages: (1) it requires no action or privacy labels during training, and (2) the anonymized videos can be directly fed into a zero-shot action recognition model, such as pretrained VLMs, without retraining. Experimental results demonstrate that our method achieves the best trade-off between privacy protection and zero-shot action recognition performance.

Our main contributions are summarized as follows:

- To the best of our knowledge, we are the first to introduce the task of privacy-preserving zero-shot action recognition, enabling PPAR methods to leverage Vision-Language Models (VLMs) for improved adaptability to novel scenes and unseen categories.

- We propose LaF-Privacy, which incorporates a new training framework that operates entirely without action or privacy labels. The anonymized videos it generates can be directly recognized by zero-shot models without retraining.

- We design a new model architecture featuring a video transformer encoder with spatial-temporal attention and an Action-Aware Masking Module that removes irrelevant patches to improve privacy while preserving action-relevant content.

- Experiments on VP-UCF101 and VP-HMDB51 datasets show that our method achieves the best trade-off between privacy protection and zero-shot action recognition, outperforming existing approaches.

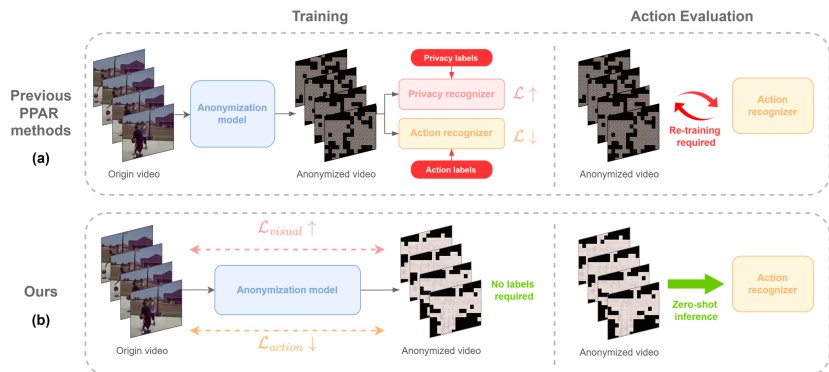

Figure 1: Comparison between previous PPAR methods and ours. (a) Prior methods require privacy/action labels and retrain action recognizers on anonymized videos. (b) Our method uses no labels and enables zero-shot inference.

## 2 RELATED WORK

### 2.1 PRIVACY-PRESERVING ACTION RECOGNITION

Privacy-preserving action recognition (PPAR) methods can be broadly categorized into three types. Downsampling-based methods (Butler et al., 2015; Ryoo et al., 2017; 2018; Srivastav et al., 2019) reduce image fidelity by lowering resolution or applying blur, to obscure sensitive details, but this severely harms action recognition due to the loss of fine-grained information. Obfuscation-based methods (Ren et al., 2018; Zhang et al., 2021; Ilic et al., 2024) use object detectors to locate sensitive regions, such as faces, and obscure them through blurring or replacement. These approaches rely on prior knowledge to specify which areas require protection and lack end-to-end trainability, making it difficult to jointly optimize privacy and action recognition, often resulting in suboptimal performance.

Recent studies have shifted toward learning-based approaches, aiming to learn transformation models that anonymize video content. Many adopt adversarial frameworks, such as (Wu et al., 2020; 2018), which jointly train an action and a privacy classifier to balance utility and privacy. BDQ (Kumawat & Nagahara, 2022) introduces a Blur, Difference, and Quantization method to achieve privacy with minimal computational cost. SPAct (Dave et al., 2022) leverages a self-supervised framework with contrastive loss to eliminate the need for privacy annotations. STPrivacy (Li et al., 2023) presents a video-level PPAR framework using vision Transformers, combining token sparsification and adversarial anonymization for effective spatio-temporal privacy removal. Peng et al. (Peng et al., 2023) apply meta-learning to enhance generalization across diverse privacy attributes and attack models. Li et al. (Li et al., 2024) replace the use of privacy labels with a distance correlation loss and introduce a patch-based attention module to perform anonymization. However, most methods still require either action or privacy labels during training and require retraining the action recognition model on anonymized videos, limiting flexibility and real-world applicability. To the best of our knowledge, no prior work achieves fully unsupervised anonymization, especially without action labels. In contrast, our proposed LaF-Privacy framework removes the need for supervision and supports zero-shot inference without retraining.

### 2.2 VISION-LANGUAGE MODEL FOR ACTION RECOGNITION

Vision-language models (VLMs) have demonstrated remarkable cross-modal generalization. CLIP (Radford et al., 2021), a pioneering work, aligns visual and textual representations in a shared embedding space through contrastive learning on large-scale image-text pairs. Its success has led to adaptations for various tasks, such as video action recognition. ActionCLIP (Wang et al., 2021) introduces the "pretrain, prompt, and fine-tune" paradigm to transfer CLIP's knowledge to video understanding. X-CLIP (Ni et al., 2022) enhances the visual encoder with a cross-frame attention mechanism, enabling the model to capture temporal dynamics in videos better. Text4Vis (Wu et al., 2023a) freezes the text encoder to extract class-level relational priors that guide the training of

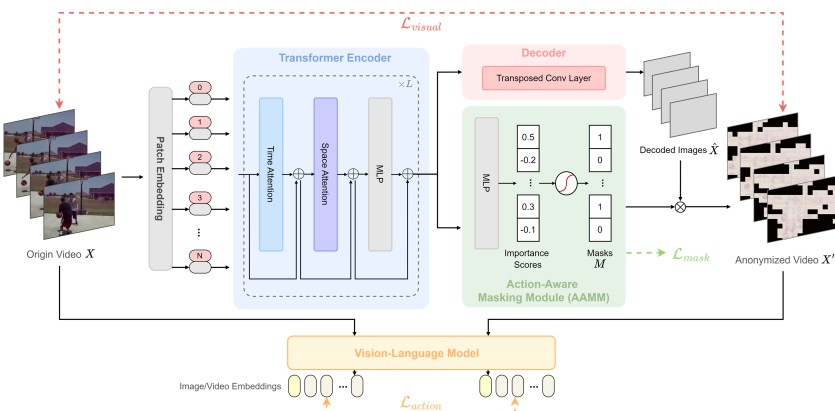

Figure 2: Proposed LaF-Privacy framework. The model takes a video input, extracts spatiotemporal features, and learns to mask unimportant regions via an Action-Aware Masking Module (AAMM). The masked output is decoded into an anonymized video, optimized to differ visually while maintaining action-related features in the vision-language embedding space.

the visual encoder. BIKE (Wu et al., 2023b) models bidirectional interactions between vision and language to enhance textual representations and alignment. These methods demonstrate the strong potential of VLMs in action recognition, particularly in enabling zero-shot capabilities without requiring task-specific annotations. However, existing approaches focus solely on action recognition and overlook privacy concerns. In this work, we propose a framework that integrates VLM-based action recognition with privacy protection, enabling privacy-preserving zero-shot learning.

## 3 METHOD

We propose LaF-Privacy, a framework for video anonymization that preserves action-relevant information while protecting visual privacy, as illustrated in Figure 2. The architecture comprises three main components: Transformer Encoder, Action-Aware Masking Module, and Decoder. The input video is first divided into patches and embedded into tokens, which are processed by the Transformer Encoder to model temporal and spatial dependencies across frames. These learned tokens are then passed to both the Action-Aware Masking Module (AAMM) and the decoder. The AAMM masks out regions irrelevant to action understanding, further enhancing privacy protection. The decoder reconstructs the video from the tokens, and the final anonymized video is produced by applying the mask to the reconstructed output.

LaF-Privacy is trained in an unsupervised setting using two objectives: (1) preserving high-level action-relevant features between the raw and anonymized videos and (2) maximizing low-level visual differences to enforce privacy. The following sections describe each module in detail.

### 3.1 TRANSFORMER ENCODER

#### 3.1.1 VIDEO TOKENIZATION AND EMBEDDING

The model takes a raw video clip $X \in \mathbb{R}^{T \times C \times H \times W}$, where $T$ consecutive RGB frames of spatial size $H \times W$ are stacked along the temporal dimension, each with $C = 3$ channels. Each frame is first partitioned into non-overlapping patches of size $P \times P$. This yields $N = \frac{H}{P}\frac{W}{P}$ patches per frame and $NT$ patches in total. After flattening each patch into a vector, we obtain the patch matrix $\mathrm{x} \in \mathbb{R}^{NT \times CP^2}$. These tokens are then mapped into an embedding space of dimension $D$ through a learnable linear projection matrix $E \in \mathbb{R}^{CP^2 \times D}$, followed by the addition of learnable spatio-temporal positional embedding $\mathrm{e}^{pos} \in \mathbb{R}^{NT \times D}$:

$$\mathrm{z}^0 = \mathrm{x}E + \mathrm{e}^{pos},$$

The resulting embeddings $\mathrm{z}^0 \in \mathbb{R}^{NT \times D}$ are then fed into the Transformer encoder.

### 3.1.2 Divided Space-Time Transformer architecture

Our Transformer Encoder consists of $L$ encoder blocks, each employing the Divided Space-time Attention proposed in (Bertasius et al., 2021) to effectively model spatio-temporal dependencies in video data. Each block sequentially applies temporal attention, spatial attention, and a feed-forward network (MLP), with Layer Normalization and residual connections:

$$z_t^\ell = \text{MSA}_{\text{time}}(\text{LN}(z^{\ell-1})) + z^{\ell-1}$$

$$z_s^\ell = \text{MSA}_{\text{space}}(\text{LN}(z_t^\ell)) + z_t^\ell$$

$$z^\ell = \text{MLP}(\text{LN}(z_s^\ell)) + z_s^\ell.$$

Here, $\ell \in \{1, ..., L\}$ denotes the block index, $\text{MSA}_{\text{time}}$ and $\text{MSA}_{\text{space}}$ stand for multi-head self-attention over time and space respectively.

For temporal attention, given input tokens $z \in \mathbb{R}^{\text{NT} \times D}$, we reshape it into $\mathbb{R}^{N \times T \times D}$ to separate the spatial and temporal axes. For each spatial location $s$, temporal attention is computed over the $T$ frames:

$$Q_t = Z_{s,:,:}W_t^Q, K_t = Z_{s,:,:}W_t^K, V_t = Z_{s,:,:}W_t^V,$$

where $W_t^Q, W_t^K, W_t^V \in \mathbb{R}^{D \times dA}$ are learnable projection matrices, $A$ is the number of attention heads, and $d = \frac{D}{A}$ is the dimensionality per head. Each head computes attention as:

$$\text{head}_i = \text{Attn}(Q_{t,i}, K_{t,i}, V_{t,i}) = \text{softmax}(\frac{Q_{t,i}K_{t,i}}{\sqrt{d}})V_{t,i}.$$

With output projection matrix $W_t^O \in \mathbb{R}^{dA \times D}$, the full multi-head temporal attention output is:

$$\text{MSA}_{\text{time}}(Q_t, K_t, V_t) = \text{Concat}(head_1, ..., head_A)W_t^O,$$

Spatial attention is applied similarly within each frame by reshaping the sequence into $z \in \mathbb{R}^{\text{T} \times \text{N} \times D}$ and attending across spatial tokens at each time step. Finally, an MLP with two linear layers, GELU activation and dropout, refines the token representations. This structure allows the model to progressively capture temporal dynamics and spatial layouts across frames.

### 3.2 Action-Aware Masking Module (AAMM)

We design the Action-Aware Masking Module (AAMM) to selectively mask regions irrelevant to action understanding, enhancing privacy protection. Beyond removing sensitive information, masking irreversibly corrupts video content, improving robustness against reconstruction attacks. In practical scenarios, masked regions can also be excluded from transmission, reducing communication overhead.

While STPrivacy (Li et al., 2023) also adopts a token pruning mechanism to preserve privacy, it uses a fixed pruning ratio, retaining only a predetermined proportion of patches across all frames. However, the regions contributing to action recognition can vary significantly between videos. For instance, in long-shot scenes where the subject appears small, a large portion of the background can be masked without affecting recognition. In contrast, for close-up scenes, excessive pruning may discard critical information necessary for accurate action understanding. Motivated by this observation, we draw inspiration from LTP (Kim et al., 2022)'s threshold-based pruning strategy. Instead of predefining a fixed pruning ratio, our approach enables the model to learn the importance of each patch and dynamically decide which ones to retain. This adaptive selection strategy offers greater flexibility and ensures that essential information is preserved across diverse video contexts.

The learned patch tokens are first passed through a score predictor, implemented using a multilayer perceptron (MLP), which estimates the importance of each token. This results in a set of importance scores $s_i$ for each token $i$. To normalize the scores, we subtract the mean score across all tokens within the same frame:

$$\widetilde{s}_i = s_i - \frac{1}{N}\sum_{j=1}^{N} s_j$$

Where $N$ denotes the number of tokens in the frame. We then apply a fixed threshold $\theta = 0$. Tokens with normalized scores $\widetilde{s}_i > 0$, meaning those above the average importance, are retained. In contrast,

tokens with $\tilde{s}_i \leq 0$ are masked. Based on this criterion, a binary mask is generated to guide the patch-level masking for each video frame.

Since hard binary decisions are not differentiable, they cannot be directly optimized through gradient-based learning. To address this issue, we adopt a soft masking strategy (Kim et al., 2022) by computing a soft mask as follows:

$$m_i = \text{softmax}\left(\frac{\tilde{s}_i}{\tau}\right)$$

Where $\tau$ is a temperature parameter that controls the sharpness of the mask, a lower value of $\tau$ yields a distribution closer to a hard mask, while higher values encourage smoother gradients. Each $m_i$ corresponds to a patch in the frame. These values are then mapped back to their spatial locations to form a full-resolution mask $M \in \mathbb{R}^{T \times C \times H \times W}$, aligned with the original video size. This soft masking approach ensures differentiability for end-to-end training.

### 3.3 Decoder and final anonymized output

Let the final-layer token embeddings from the encoder be $z^L \in \mathbb{R}^{NT \times D}$, where each token corresponds to a patch in the video frames. The tokens are passed through a transposed convolution layer $D(\cdot)$ to reconstruct the spatial patches: $\hat{X} = D(z^L)$. To generate the anonymized video, we apply an element-wise multiplication between the reconstructed frames $\hat{X}$ and the mask $M$ produced by AAMM, resulting in the final anonymized video $X' = \hat{X} \odot M$.

### 3.4 Training objectives

#### 3.4.1 Maximizing visual difference

To ensure privacy protection, we guide the model in generating anonymized videos that are visually distinct from the original input. This is achieved by maximizing the difference between the original and anonymized frames. We adopt a combination of masked Mean Squared Error (MSE) and Structural Similarity Index Measure (SSIM) as our visual loss:

$$\mathcal{L}_{visual} = \lambda_{mse} \cdot \text{MSE}_{\text{mask}}(X, X', M) + \lambda_{ssim} \cdot (1 - \text{SSIM}(X, X'))$$

Here, $M$ is a binary mask where $M_{i,j} = 1$ indicates unmasked regions determined by AAMM. The masked MSE is defined as:

$$\text{MSE}_{\text{mask}}(X, X', M) = \frac{1}{\sum M} \sum_{i,j} M_{i,j} \cdot (X_{i,j} - X'_{i,j})^2$$

It aligns with the masking result by focusing only on the remaining patches. MSE penalizes direct pixel-wise similarity, encouraging large low-level differences, which is effective for removing fine-grained visual details. However, it does not align well with human perception. SSIM, on the other hand, captures perceptual aspects such as luminance, contrast, and structural consistency. By combining both, the model learns to obscure sensitive information not only at the pixel level but also in a way that is perceptually noticeable to humans, leading to stronger privacy preservation.

#### 3.4.2 Preserving action features

To ensure that the anonymized video retains its utility for action recognition, we encourage the visual-language model (VLM) embeddings of the original and anonymized videos to remain close. $X$ and $X'$ denote the original and anonymized videos, respectively. Both are processed through a pretrained video-based VLM, where each frame is divided into patches and passed through the visual encoder, producing patch embeddings $y^{patch}$ and a [CLS] token embedding $y^{cls}$ per frame. In video-level modeling, the sequence of $T$ [CLS] embeddings is further processed to capture cross-frame dependencies and generate the final video embedding $y^{video}$. The video embedding is then compared to text embeddings to perform action classification. To align the representations between $X$ and $X'$, we compute the Mean Squared Error (MSE) between their corresponding embeddings:

$$\mathcal{L}_{action} = \text{MSE}(y_X^{patch}, y_{X'}^{patch}) + \text{MSE}(y_X^{cls}, y_{X'}^{cls}) + \text{MSE}(y_X^{video}, y_{X'}^{video})$$

This embedding-level supervision ensures that the anonymized video $X'$ remains semantically aligned with the original $X$ in the VLM feature space, preserving its action recognition performance.

### 3.4.3 MASKING REGULARIZATION

To encourage the AAMM to mask out as many uninformative or privacy-sensitive patches as possible, we introduce a regularization term that penalizes the number of retained tokens. Specifically, we apply an L1 loss over the mask values:

$$\mathcal{L}_{mask} = \frac{1}{THW} \sum_{t=1}^{T} \sum_{i=1}^{HW} \|M_{t,i}\|_1$$

### 3.4.4 OVERALL OBJECTIVES

The final training objective combines the three loss components described above: visual loss, action loss, and masking regularization. Since the visual loss is designed to maximize the perceptual difference between the original and anonymized videos, we assign it a negative weight. The overall loss is defined as:

$$\mathcal{L}_{total} = \lambda_{action} \cdot \mathcal{L}_{action} - \lambda_{vis} \cdot \mathcal{L}_{visual} + \lambda_{mask} \cdot \mathcal{L}_{mask}$$

The loss weights control the trade-off between privacy protection and action recognition performance.

## 4 EXPERIMENTS

### 4.1 DATASETS

We conducted experiments on VP-HMDB51 and VP-UCF101 datasets (Li et al., 2023). VP-HMDB51 extends the HMDB51 (Kuehne et al., 2011) dataset, consisting of 6,849 videos spanning 51 categories of human actions. Similarly, VP-UCF101 is an extension of the UCF101 (Soomro et al., 2012) dataset, comprising 13,320 videos across 101 types of sports actions. Both datasets include action and privacy annotations for each video. In our setting, these labels are only used for evaluation and remain unseen during training.

Action labels specify the type of action performed in the video. Privacy labels cover five types of privacy attributes: face, skin color, gender, nudity, and relationships. Each privacy label is a binary indicator that reflects whether the corresponding attribute is identifiable throughout the video.

### 4.2 EVALUATION METRICS

We adopted single-view evaluation on the test split of each dataset. For action evaluation, anonymized videos are directly fed into pretrained VLMs in a zero-shot setting. Higher classification accuracy indicates higher utility. For privacy evaluation, following STPrivacy (Li et al., 2023), we train a ViT-based privacy recognizer with anonymized videos and their corresponding privacy labels. This is formulated as a multi-label binary classification task. The model's performance is assessed using cMAP and class-wise F1 score. Lower values in both metrics indicate better privacy preservation.

### 4.3 IMPLEMENTATION DETAILS

We adopted the ViT-B transformer with a patch size of 16 for both LaF-Privacy and evaluation models. Video clips are of size 8×224×224. The frame sampling rate is set to 4 for VP-UCF101 and 2 for VP-HMDB51. The transformer encoder uses L=12 layers, and $\tau$ for AAMM is set to 0.05. Experiments are conducted with X-CLIP (Ni et al., 2022) and ActionCLIP (Wang et al., 2021) as the VLM. Loss weights are set as follows: $\lambda_{mse} = 1, \lambda_{ssim} = 1, \lambda_{action} = 1; \lambda_{mask}$ is 1 for VP-HMDB51 and 2 for VP-UCF101. Further details are shown in the appendices. The model is first pretrained on reconstructing the images. Optimization is performed using SGD. All experiments are conducted on two NVIDIA RTX A5000 GPUs.

### 4.4 COMPARISON ANALYSIS

Since there are currently no existing works addressing zero-shot PPAR, we compare our method with several unsupervised baseline approaches following (Dave et al., 2022). We also re-implemented

Table 1: Experiment results using X-CLIP (Ni et al., 2022) as the VLM model. The best results are highlighted in bold, while the second-best results are underlined.

| Setting | Method | VP-HMDB51 | | | VP-UCF101 | | |
|---|---|---|---|---|---|---|---|
| | | Action Top1 ↑ | Privacy F1 ↓ | Privacy cMAP ↓ | Action Top1 ↑ | Privacy F1 ↓ | Privacy cMAP ↓ |
| Full-shot | Raw | 69.99 | 0.6819 | 0.7043 | 85.38 | 0.668 | 0.6979 |
| | STPrivacy | 65.79 | 0.5329 | 0.619 | 75.63 | 0.5366 | 0.6219 |
| Zero-shot | Raw | 42.352 | 0.6819 | 0.7043 | 73.249 | 0.668 | 0.6979 |
| | STPrivacy | 1.208 | 0.5329 | 0.619 | 1.507 | 0.5366 | 0.6219 |
| | 2× down | **41.458** | 0.6835 | 0.7014 | **69.574** | 0.6661 | 0.6961 |
| | 4× down | 32.469 | 0.6747 | 0.6996 | 56.172 | 0.6677 | 0.6958 |
| | Blackening | 21.154 | **0.6353** | **0.6686** | 50.806 | **0.6417** | **0.6791** |
| | Strong blur | 29.159 | 0.6674 | 0.6944 | 59.529 | 0.6637 | 0.6935 |
| | Weak blur | 34.392 | 0.6699 | 0.6936 | 64.420 | 0.6607 | 0.6937 |
| | Ours | 38.998 | 0.632 | 0.6717 | 65.239 | **0.6417** | **0.6791** |

Table 2: Results obtained in the ablation studies.

| | Method | Action Top1 ↑ | Privacy F1 ↓ | Privacy cMAP ↓ | | Method | Action Top1 ↑ | Privacy F1 ↓ | Privacy cMAP ↓ |
|---|---|---|---|---|---|---|---|---|---|
| Visual losses | SSIM | 34.079 | 0.6423 | 0.6812 | Action losses | w/o vid | 35.242 | 0.6355 | 0.6793 |
| | L1+SSIM | 36.762 | 0.6389 | 0.6807 | | w/o cls | 36.986 | 0.6321 | 0.6869 |
| | MSE | 38.193 | 0.6429 | 0.6865 | | w/o patch | 37.835 | 0.6413 | 0.6843 |
| | MSE+SSIM (ours) | **38.998** | **0.632** | **0.6717** | | vid+cls+patch (ours) | **38.998** | **0.632** | **0.6717** |
| Applying mask | Unmasked | 38.417 | 0.6335 | 0.6821 | | | | | |
| | Masked (ours) | **38.998** | **0.632** | **0.6717** | | | | | |

STPrivacy (Li et al., 2023), the current full-shot SOTA method, based on descriptions in the paper, since only partial code was available. Other prior works are full-shot methods and not the focus of our comparison. Their performance is also expected to be inferior to STPrivacy, and thus are excluded.

Table 1 presents the results using X-CLIP (Ni et al., 2022) as the VLM. We also show full-shot results with anonymized videos and labels trained on a ViT-B recognizer. Among zero-shot methods, the 2× downsampling achieves the best action recognition accuracy due to its minimal blurring, but offers almost no privacy protection. In contrast, the blackening method, which masks the identified regions directly, achieves the best privacy protection but at the cost of significantly degraded action recognition performance. Our method achieves either the best or second-best performance across both action and privacy metrics, and overall provides the best trade-off between utility and privacy. While STPrivacy performs well in a fully supervised setting, it fails under zero-shot conditions due to the severe degradation of video quality, making it impossible for the zero-shot model to perform predictions. Furthermore, STPrivacy requires access to both action and privacy labels for all videos during training the anonymization model, whereas our method operates entirely without any supervision.

We evaluate privacy preservation using F1 score and cMAP. Although metric differences may appear small in scale, they should be interpreted in context. Consider a random predictor that assigns equal probabilities (0.5) to both classes, and assume the proportion of positive samples in the dataset is $p$. The expected F1 score would be $p/(0.5 + p)$, and the expected cMAP would be $p$. For VP-HMDB51 and VP-UCF101, where $p = 0.583$ and $p = 0.587$, the lower bounds for F1 score are approximately $0.538$ and $0.54$, respectively.

We further evaluate our method using another VLM, ActionCLIP (Wang et al., 2021), with results in the appendices. Our method consistently achieves the best trade-off across metrics, showing good generalization to different VLMs. We also conduct a cross-VLM evaluation, showing that even when different VLMs are used for training and testing, the model maintains a reasonable recognition performance. The cross-VLM results and additional visualizations are provided in the appendices.

## 4.5 ABLATION STUDIES

We conduct ablation studies to validate the effectiveness of our framework. All experiments in this section are performed on the VP-HMDB51 dataset using X-CLIP (Ni et al., 2022) as the VLM. More ablation results can be found in the appendices.

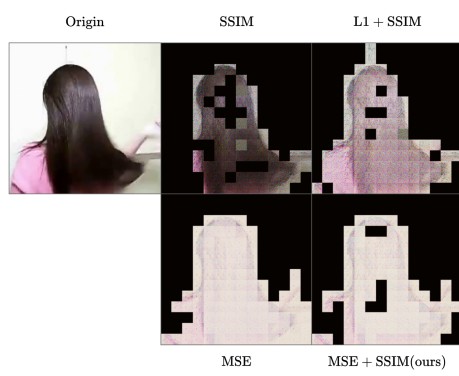

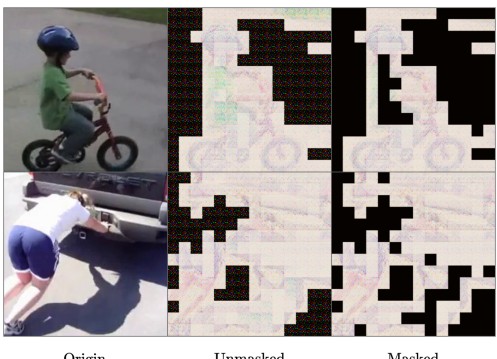

Figure 3: Visualization results of different visual losses.

Figure 4: Visualization results with and without the mask from AAMM.

### 4.5.1 VISUAL LOSSES

We evaluate different visual losses designed to increase the perceptual distance between the original and anonymized videos. As shown in Table 2, our method uses a combination of MSE and SSIM losses. Removing either component or replacing MSE with L1 loss leads to a drop in privacy protection, without improving action recognition performance. The visualizations in Figure 3 further illustrate these differences. When using only SSIM, the anonymized frames retain much of the original color and structure, making privacy protection less effective. Adding L1 introduces more pixel differences, but still performs worse than MSE. Incorporating SSIM introduces local distortions and noise, helping to remove texture and fine details, which improves the anonymization effect.

### 4.5.2 ACTION LOSSES

To preserve action-relevant features, we minimize the distance between the original and anonymized VLM embeddings, including video-level, CLS (frame-level), and patch-level representations. As shown in Table 2, removing video-level embeddings results in the largest performance drop, followed by CLS and patch embeddings. This indicates that higher-level embeddings contribute more significantly, and using all three types yields the best overall performance.

### 4.5.3 ACTION-AWARE MASKING

As shown in Figure 4, even before applying the AAMM, the transformer encoder has already learned to suppress certain uninformative patches, which appear as darker regions. The AAMM further masks these patches, introducing stronger visual degradation. Guided by masking regularization, it encourages masking more patches while preserving action-relevant features during training. Quantitative results in Table 2 show that applying the mask does not degrade action recognition performance and even improves privacy metrics (lower F1 and cMAP), indicating improved privacy protection.

## 5 CONCLUSION

In this paper, we propose LaF-Privacy, a label-free framework for PPAR that requires no action or privacy annotations during training. The anonymized videos produced by our model can be directly used with pretrained VLMs for zero-shot action recognition, eliminating the need for retraining and significantly improving flexibility and practicality compared to previous methods. Our approach leverages a transformer encoder to capture spatio-temporal dynamics and an Action-Aware Masking Module to remove irrelevant patches, further enhancing privacy protection. Experimental results demonstrate that LaF-Privacy achieves the best trade-off between action recognition accuracy and privacy preservation. A current limitation of our method is the slight performance drop when different VLMs are used for training and testing. Future work may explore strategies to improve cross-VLM generalization, enabling broader applicability in diverse deployment scenarios.

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

APPENDICES

# A  TRAINING PROCEDURE

To stabilize optimization and guide the model toward the desired objectives, we adopt a three-phase training strategy:

**Phase 1: visual + action supervision.** We begin by training only with visual loss and action loss. This allows the model to learn how to maximize visual difference while preserving action-relevant features. During this phase, the model implicitly learns which patches are essential for recognition.

**Phase 2: masking optimization.** We then introduce the AAMM and enable the masking regularization loss while continuing to supervise action consistency. To ensure training stability, the visual loss is temporarily removed.

**Phase 3: full objective fine-tuning.** Once masking behavior stabilizes, we reintroduce the visual loss to fine-tune the model, ensuring effective anonymization while retaining task performance.

## A.1  IMPLEMENTATION DETAILS

For the loss weights, $\lambda_{vis}$ is 1 in Phase 1, and [0.5, 0.2] for VP-HMDB51 and VP-UCF101 in Phase 3. Training runs for [60, 20, 40] epochs with learning rates [1e-3, 1e-3, 5e-4] in the three phases.

# B  MORE EXPERIMENT RESULTS

## B.1  EXPERIMENT RESULTS ON USING ACTIONCLIP AS VLM

We further evaluate our method using another VLM, ActionCLIP (Wang et al., 2021), with the results shown in Table 3. Similar results are observed that our method consistently achieves the best trade-off across all metrics, demonstrating that our framework generalizes well to different VLM-based action recognition models.

## B.2  CROSS-VLM EVALUATION

Table 4 presents the top-1 action recognition accuracy across different Vision-Language Models. When different VLMs are used for training and testing, the performance remains reasonably high, though slightly lower compared to the setting where the same VLM is used for both. This suggests that while embeddings from different VLMs are inherently similar, subtle differences still exist. Future work may explore strategies to improve generalization across diverse VLMs.

## B.3  ABLATION STUDY ON TRAINING STRATEGIES

We adopt a three-phase training strategy to improve training stability. As shown in Table 5, training with only phase 3, where all objectives are optimized from the beginning, results in unstable convergence. This makes it difficult for the action loss to decrease and leads to poor recognition performance. Similarly, skipping phase 2 and transitioning directly from phase 1 to phase 3 causes the visual loss to interfere with masking, which prevents the action loss from staying low. These results highlight the importance of our progressive training design.

## B.4  ABLATION STUDY ON MASKING LOSS

In our Action-Aware Masking Module, we adopt threshold-based masking and L1 loss to regularize the masking process. In contrast, STPrivacy (Li et al., 2023) introduces a regularization loss to enforce the preserved region proportion to match a predefined ratio $\alpha$, formulated as:

$$\mathcal{L}_{mask} = \frac{1}{T} \sum_{t=1}^{T} \left( \frac{1}{HW} \sum_{i=1}^{HW} M_{t,i} - \alpha \right)^2$$

Table 3: Experiment results using ActionCLIP (Wang et al., 2021) as the VLM model.

| Setting | Method | VP-HMDB51 | | | VP-UCF101 | | |
|---------|--------|-----------|--|--|-----------|--|--|
| | | Action Top1 ↑ | Privacy F1 ↓ | Privacy cMAP ↓ | Action Top1 ↑ | Privacy F1 ↓ | Privacy cMAP ↓ |
| Full-shot | Raw | 69.99 | 0.6819 | 0.7043 | 85.38 | 0.668 | 0.6979 |
| | STPrivacy | 65.79 | 0.5329 | 0.619 | 75.63 | 0.5366 | 0.6219 |
| Zero-shot | Raw | 47.151 | 0.6819 | 0.7043 | 69.227 | 0.668 | 0.6979 |
| | STPrivacy | 1.884 | 0.5329 | 0.619 | 1.192 | 0.5366 | 0.6219 |
| | 2× down | **45.772** | 0.6835 | 0.7014 | **66.34** | 0.6661 | 0.6961 |
| | 4× down | 34.421 | 0.6747 | 0.6996 | 47.458 | 0.6677 | 0.6958 |
| | Blackening | 22.518 | **0.6353** | **0.6686** | 47.246 | 0.6417 | 0.6791 |
| | Strong blur | 29.6415 | 0.6674 | 0.6944 | 55.588 | 0.6637 | 0.6935 |
| | Weak blur | 36.672 | 0.6699 | 0.6936 | 59.216 | 0.6607 | 0.6937 |
| | Ours | 42.371 | 0.6555 | 0.6864 | 61.388 | **0.616** | **0.6684** |

Table 4: Top-1 action recognition accuracy under cross-VLM settings.

| | | HMDB51 | | UCF101 | |
|--|--|--------|--|--------|--|
| | | Test | | | |
| | | X-CLIP | ActionCLIP | X-CLIP | ActionCLIP |
| Train | X-CLIP | 38.998 | 39.568 | 65.239 | 58.872 |
| | ActionCLIP | 35.107 | 42.371 | 56.172 | 61.388 |

Table 5: Ablation study on different training strategies.

| Method | Action Top1 ↑ | Privacy F1 ↓ | Privacy cMAP ↓ |
|--------|---------------|--------------|----------------|
| Phase 1+3 | 36.852 | 0.6481 | 0.6885 |
| Only phase 3 | 34.615 | **0.6226** | 0.6793 |
| Phase 1+2+3 (ours) | **38.998** | 0.632 | **0.6717** |

They set $\alpha = 0.7$ and gradually apply the masking three times, resulting in a final preserved ratio of $0.7^3$ patches in the anonymized output. We conduct an ablation study by replacing our masking loss with the one from STPrivacy and experimenting with different preservation ratios. As shown in Table 6, a lower preservation ratio leads to more regions being masked, which enhances privacy protection. When $\alpha = 0.7$, the privacy-preserving performance is comparable to ours. However, enforcing a fixed preservation ratio across all video frames harms action recognition accuracy. Specifically, lower preservation ratios result in decreased action performance.

Visualization examples are shown in Figure 5. It can be observed that with the fixed-ratio loss, the model fails to mask the background regions (e.g., black areas) identified by the Transformer Encoder and instead masks more informative regions, reducing visual context. For instance, in (a), a video of a person *diving* from a bridge, fixed-ratio methods excessively mask the bridge, leading to incorrect predictions such as *somersault*. In contrast, our method preserves the bridge area, enabling correct action recognition. Similarly, in (b), for a *climbing stairs* video, fixed-ratio methods mask the stairs, resulting in misclassification as *walking*. These results suggest that selectively masking frames based on learned features, rather than enforcing a fixed ratio, leads to more flexible and effective performance in both privacy protection and action recognition.

Table 6: Ablation study on different masking losses.

| Method | Action Top1 ↑ | Privacy F1 ↓ | Privacy cMAP ↓ |
|---|---|---|---|
| Fixed ratio=0.5 | 37.343 | **0.6211** | **0.6694** |
| Fixed ratio=0.7 | 37.657 | 0.6337 | 0.6718 |
| Non-fixed ratio (ours) | **38.998** | 0.632 | 0.6717 |

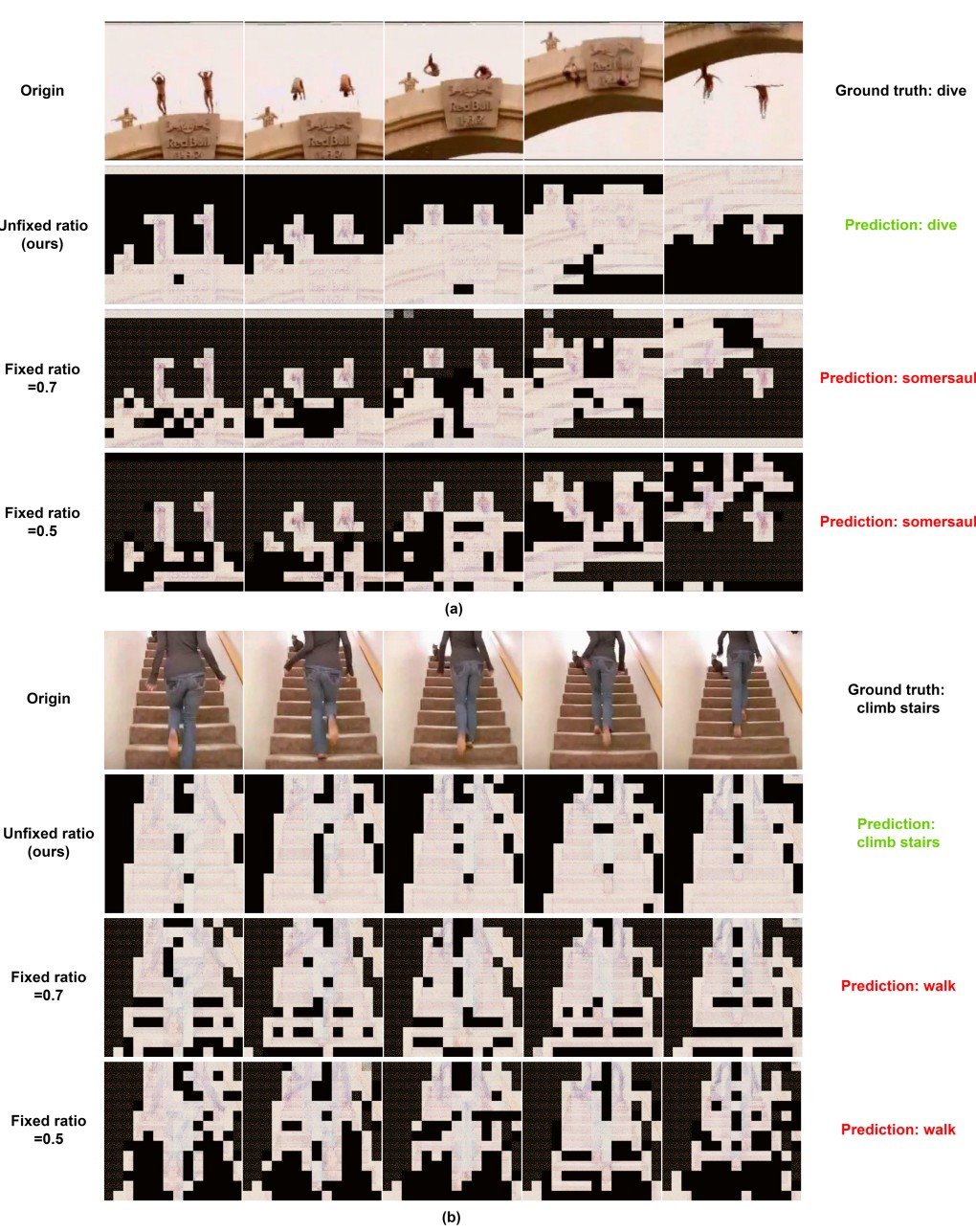

Figure 5: Visualization results of different masking losses.

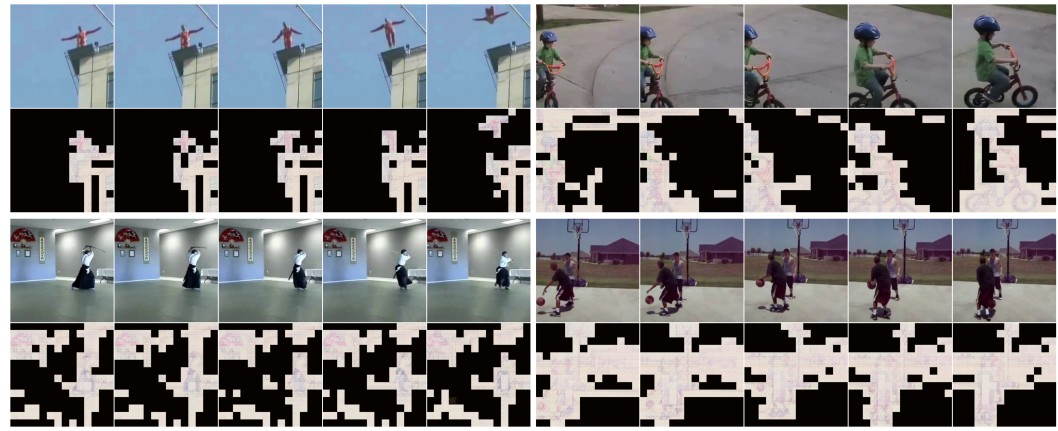

Figure 6: Visualization results of the anonymized videos on VP-HMDB51 dataset.

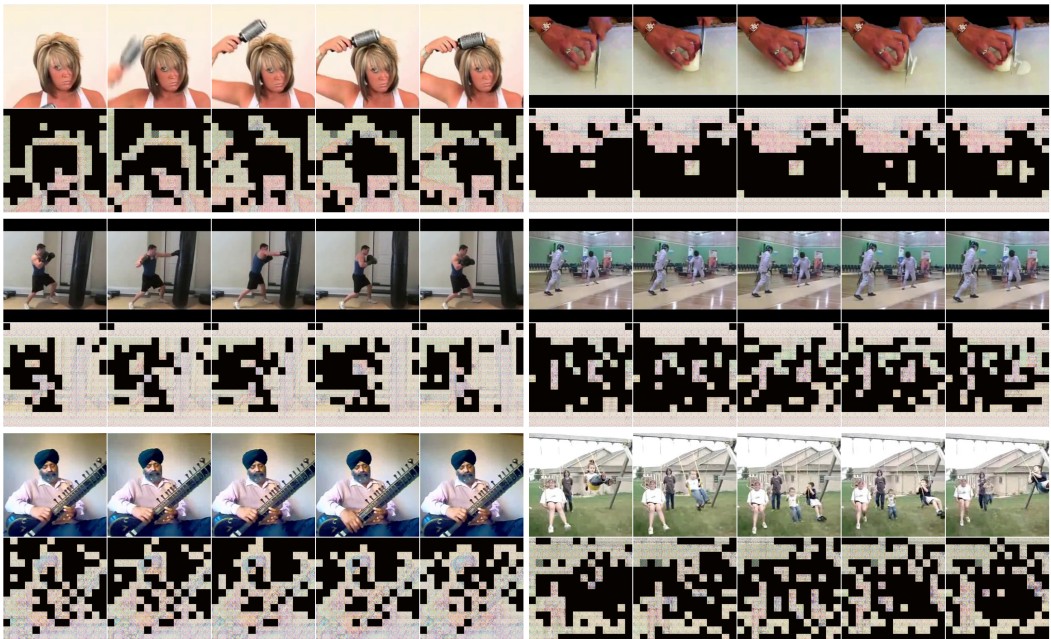

Figure 7: Visualization results of the anonymized videos on VP-UCF101 dataset.

## B.5 VISUALIZATION RESULTS

Figure 6 and 7 present additional visualizations of the anonymized videos on the VP-HMDB51 and VP-UCF101 datasets. It can be observed that action-irrelevant regions, such as the background, are masked by the AAMM. Moreover, the remaining visible areas are visually indistinct, effectively protecting privacy while preserving action-relevant features.

Table 7: Number of parameters of the models.

|  | STPrivacy | Ours |
|---|---|---|
| Num of parameter | 200M | 122M |

Table 8: Prediction performance of different privacy attributes.

|  | Positive sample ratios | F1 | cMAP | F1 Gain | cMAP Gain |
|---|---|---|---|---|---|
| Face | 0.91 | 0.5925 | 0.9192 | -0.0519 | 0.0132 |
| Skin color | 0.62 | 0.7129 | 0.7426 | 0.1601 | 0.1246 |
| Gender | 0.92 | 0.5314 | 0.9249 | -0.1165 | 0.0049 |
| Nudity | 0.26 | 0.6747 | 0.4167 | 0.3343 | 0.1587 |
| Relationships | 0.212 | 0.6526 | 0.3547 | 0.3548 | 0.1427 |

## B.6 COMPUTATION COSTS

Table 7 shows the number of parameters of the models, both based on the ViT-B structure. Since our framework does not include the action and privacy recognition models, the training parameters are much lower. In contrast, STPrivacy (Li et al., 2023) jointly trains both recognizers, resulting in higher computational cost. All experiments are conducted on 2 NVIDIA RTX A5000 GPUs, each with 24 GB of memory. The full training time approximately takes 24 hours for the VP-HMDB51 dataset and 48 hours for the VP-UCF101 dataset.

## B.7 PERFORMANCE OF EACH PRIVACY ATTRIBUTE

Table 8 presents the positive sample ratios for different privacy attributes, along with their corresponding recognition performance. Due to the imbalanced class distributions across attributes, direct comparison of F1 and cMAP scores can be misleading. To address this, we incorporate F1/cMAP Gain, which measures the difference between the model's actual performance and that of a random predictor baseline. A lower gain indicates stronger privacy preservation.

Among all attributes, gender and face exhibit the strongest privacy protection, likely because these categories rely heavily on fine-grained texture and facial features that are effectively obscured through anonymization. In contrast, skin color and nudity are less protected, as they depend on broader color distributions and human body contours, features that may still be partially visible even after privacy processing. Relationships show the weakest protection. This can be attributed to the fact that relational cues often stem from spatial arrangements rather than detailed appearance, and such high-level layout information tends to persist even after visual obfuscation, making it difficult to fully conceal.

