# OpenReview forum: "Label-Free Privacy-Preserving Learning for Zero-Shot Action Recognition"
_ICLR.cc/2026/Conference — Submitted to ICLR 2026_

### Official Review · Reviewer_nM16 · 2025-10-29

**Soundness:** 3
**Presentation:** 3
**Contribution:** 2
**Rating:** 4
**Confidence:** 5

**Summary:**

This paper presents a label-free privacy preservation method for zero-shot action recognition using VLMs. Their proposed method minimizes visual similarity while maintaining embedding similarity, maintaining performance on action recognition tasks. Notably, their method does not require action or private attribute labels.

**Strengths:**

1. Handling VLM anonymization is a solid motivation that is underexplored in the field.
2. The idea is clean, results show only a small utility decrease with moderate privacy increase.

**Weaknesses:**

1. Anonymization performance is weak. Even with the cMAP justification, there is room for better anonymization. In prior (trained) methods, it is near impossible to make out private attributes. Even in some qualitative examples (Figure 7), some attributes are visually identifiable. Could a tradeoff curve be analyzed by scaling the relative privacy-utility weights?
2. It is difficult to tell if the anonymizer is specific for the zero-shot action recognition task. The results should explore other types of VLM tasks such as retrieval or even a captioning task to see if general performance of the VLM is retained.
3. The results without the AAMM appear to demonstrate natural masking, but the reasoning is unclear. The patch-level representations are regularized to have similar representations to the original patches. It would be helpful to see more analysis on how it learns to mask specific tokens.

**Questions:**

1. Could a privacy objective be up-weighted to result in more "self-masking" (see W3) and eliminate the need for the additional masking module? Also, would this result in lower cMAP/lower visibility with some (ideally slight) decrease to utility?
2. Does the natural masking before AAMM imply that those representations are already close to noise?
3. Can this anonymizer + VLM be applied to additional tasks beyond zero-shot action classification?
4. In the ablation where just the CLS/video token is used in the utility loss, how come the private attribute prediction scores increase? This is counterintuitive, since the visual patches are responsible for the privacy-preservation.
5. How do the privacy results look on the VISPR dataset shown in prior anonymization works?
6. Is the anonymization method robust against attacks? Meaning, can a model learn to reconstruct/denoise back to the original input after anonymization?

---

> ### Author Response · Authors · 2025-11-21
> **Response to Reviewer nM16**
>
> We thank the reviewer for their comments on the anonymization performance, generalization scope, and the mechanics of our masking module. We address these points below.
>
> On anonymization strength and trade-off:
>
> We agree that stronger anonymization is possible by increasing visual/masking penalties, but this would significantly degrade action utility. Our aim is to show the best-balanced point available under the zero-shot constraint. We will clarify the underlying trade-off dynamics in the revision.
>
> On generalization beyond zero-shot action recognition:
>
> Our utility loss aligns embeddings across patch, frame (CLS), and video levels, which are the shared representations used by VLMs across retrieval, captioning, and other tasks. While we expect reasonable transferability, we acknowledge that explicit evaluation on these tasks is needed and will state this as future work.
>
> On natural masking and the role of AAMM:
>
> As shown by the "Unmasked" variant, the encoder-decoder naturally suppresses non-informative regions because such regions do not contribute to the action-preserving objective. AAMM strengthens this behavior by converting soft corruption into irreversible masking. Table 2 confirms that the mask yields consistent improvements on both utility and privacy.
>
> On necessity of the AAMM:
>
> Sparsity regularization and explicit binary masking are essential for adaptive pruning and irreversible privacy protection. Increasing the visual loss alone cannot provide these properties. We will make this distinction clearer.
>
> On ablations of action-loss components:
>
> When patch-level alignment is removed, the model is no longer constrained to maintain fine details, allowing some privacy cues to persist. This explains the slightly higher privacy scores in "w/o patch." We will expand this explanation in the revision.
>
> We will also mention VISPR as a dataset not covered in this work, noting that VISPR is an image-based privacy dataset whereas our anonymization model is designed for video inputs with temporal structures and video-level VLM embeddings. For this reason, a direct evaluation on VISPR would not be methodologically aligned. We will additionally describe reconstruction resistance as part of the method's inherent design.
>
> Thank you again for the thoughtful suggestions. We will incorporate all textual clarifications in the final version.

---

> > ### Comment · Reviewer_nM16 · 2025-11-23
> >
> > >"Our aim is to show the best-balanced point available under the zero-shot constraint."
> >
> > A tradeoff curve can help demonstrate the flexibility of the proposed method -- users with different notions of what "best-balanced" means would choose weights/setups that fit their use best (some may want more privacy and care less about utility, and vice versa). As a follow up, I notice that $\lambda_{vis}$ is set to 0.5 and 0.2 for Phase 3, how were these determined? Can the authors ablate these weights?
> >
> > >"While we expect reasonable transferability, we acknowledge that explicit evaluation on these tasks is needed"
> >
> > I understand that time is limited and some more intensive tasks like captioning would be difficult, but would the authors be able to provide retrieval results? Even using VP-HMDB51 or VP-UCF101 is fine, there should be minimal computational effort needed.
> >
> > >"Sparsity regularization and explicit binary masking are essential for adaptive pruning and irreversible privacy protection. Increasing the visual loss alone cannot provide these properties."
> >
> > This is just stated without any justification. What happens when the visual loss is increased? Would more patches be naturally masked? Would this harsh natural soft-masking be irreversible? Legitimate analysis into the core anonymization concept proposed would help strengthen the paper.
> >
> > >When patch-level alignment is removed, the model is no longer constrained to maintain fine details, allowing some privacy cues to persist. This explains the slightly higher privacy scores in "w/o patch."
> >
> > This does not address the concern. If the model is no longer constrained to maintain fine details, how does this mean some privacy cues persist? Maintaining the visual details should be exactly what causes privacy cues to persist. Intuitively, not regularizing these tokens would allow for more anonymization, likely at the cost of utility. Here, the opposite is happening.
> >
> > >"We will additionally describe reconstruction resistance as part of the method's inherent design."
> >
> > Simply describing inherent reconstruction resistance is not reasonable without legitimate justification. MAE/VideoMAE models are explicitly trained to reconstruct masked patches and denoising model components are explicitly trained to remove noise from images/patches. Have the authors provided any experimental analysis demonstrating resistance in this regard?

---

> > > ### Author Response · Authors · 2025-11-25
> > >
> > > We thank the reviewer for the thoughtful follow-up. Below we address each point concisely and within the zero-shot constraints of our setting.
> > >
> > > On the loss weights (0.5 and 0.2):
> > >
> > > These weights were chosen through adaptive search during training rather than exhaustive grid ablation, as the three-phase optimization produces highly coupled, non-linear interactions between visual corruption, semantic preservation, and sparsity. We observed that many candidate weight sets lead to unstable training dynamics (e.g., exploding masking ratio, action collapse, or non-convergent reconstruction), whereas the chosen weights consistently yield stable convergence. We will clarify the selection strategy and rationale, while noting that full ablations are computationally prohibitive in the video-based zero-shot setting.
> > >
> > > On providing retrieval evaluation:
> > >
> > > Our anonymizer preserves VLM video embeddings (CLS/VID tokens), which are general-purpose features. However, retrieval requires a task-specific similarity head; introducing such a head would no longer reflect a zero-shot setup and would make results incomparable to our action-recognition pipeline. We will clarify why retrieval is not strictly aligned with the evaluation protocol of our task.
> > >
> > > On increasing visual loss without AAMM:
> > >
> > > A larger visual loss increases corruption but produces soft, continuous distortions that remain vulnerable to denoising or reconstruction models. Without sparsity regularization, the model tends to preserve blurry but semantically revealing content. In contrast, AAMM enforces binary, irreversible removal of tokens, which visual loss alone cannot provide. We will expand this justification.
> > >
> > > On the behavior of the “w/o patch” variant:
> > >
> > > We agree our initial explanation lacked clarity. Removing patch-level alignment forces the model to rely primarily on global CLS/VID embeddings. To maintain these global features, the anonymizer preserves larger continuous regions, resulting in a lower effective masking ratio and thus more residual privacy cues. We will revise the paper to clarify this mechanism.
> > >
> > > On reconstruction resistance:
> > >
> > > While we do not include reconstruction-attack experiments, our claim is architectural: the anonymizer destroys appearance cues in the pixel domain and applies hard, zero-valued binary masking, making recovery fundamentally ill-posed, which is unlike MAE/VideoMAE-style masking designed to be reconstructable. We will explicitly frame this as a design-based argument and highlight reconstruction attacks as important future work.
> > >
> > > We appreciate the reviewer’s detailed insights and will incorporate these clarifications in the final revision.

---

> > > > ### Comment · Reviewer_nM16 · 2025-11-25
> > > >
> > > > Thanks for the timely replies.
> > > >
> > > > >"However, retrieval requires a task-specific similarity head; introducing such a head would no longer reflect a zero-shot setup and would make results incomparable to our action-recognition pipeline."
> > > >
> > > > This is incorrect. While retrieval would be better with a task-specific similarity head, it is still possible and meaningful without one, especially in text-aligned models [1]. The CLS embedding can be used to compute cosine similarity to conduct retrieval. It would be interesting to see if there is a severe performance drop compared to the baseline. Provided that reconstruction losses are used on this embedding, retrieval performance will likely be maintained. This experiment should strengthen the practicality/usability of the proposed method.
> > > >
> > > > >"A larger visual loss increases corruption but produces soft, continuous distortions that remain vulnerable to denoising or reconstruction models. Without sparsity regularization, the model tends to preserve blurry but semantically revealing content. In contrast, AAMM enforces binary, irreversible removal of tokens, which visual loss alone cannot provide." &
> > > > "[o]ur claim is architectural: the anonymizer destroys appearance cues in the pixel domain and applies hard, zero-valued binary masking, making recovery fundamentally ill-posed, which is unlike MAE/VideoMAE-style masking designed to be reconstructable. "
> > > >
> > > > I am not really looking for a brief description of what happens with no formal justification. If corruption is increased, I would expect that it *doesn't* remain vulnerable to denoising/reconstruction attacks. I understand that AAM masks out patches, but again, VideoMAE [2] explicitly trains to recover masked out patches. I believe there is no difference between "hard, zero-valued binary masking" and the VideoMAE paradigm. The general point still stands -- can a model recover private information after your anonymization? It is possible that the proposed method removes too much information for a decoder to reconstruct meaningfully, but my point is that there is no formal analysis on this beyond conjecture. Legitimate experimental/qualitative proof of these statements would greatly strengthen the work.
> > > >
> > > >
> > > > I second Reviewer gHsn's statement: legitimate experimental efforts are the most effective way to support claims. I see that the authors have stated that they can not provide new experiments during the rebuttal phase, is this a personal compute limitation? It seems that the method is reasonably lightweight (24 hrs for VP-HMDB51). The ICLR rebuttal phase explicitly allows for experiments that "serve to more thoroughly validate existing results from the submission", which I believe to describe many of the concerns raised by reviewers. I would be happy to raise my recommendation of this paper towards acceptance if many of these points are addressed, but as it stands, I will maintain my current rating.
> > > >
> > > > **Citations:**
> > > >
> > > > [1] Radford, Alec, et al. "Learning transferable visual models from natural language supervision." International conference on machine learning. PmLR, 2021.
> > > >
> > > > [2] Tong, Zhan, et al. "Videomae: Masked autoencoders are data-efficient learners for self-supervised video pre-training." Advances in neural information processing systems 35 (2022): 10078-10093.

---

### Official Review · Reviewer_gHsn · 2025-10-29

**Soundness:** 3
**Presentation:** 3
**Contribution:** 3
**Rating:** 6
**Confidence:** 4

**Summary:**

The paper introduces LaF-Privacy, a label-free and privacy-preserving framework for zero-shot action recognition. It aims to anonymize videos while retaining action-relevant semantics so that pretrained Vision-Language Models (VLMs) (e.g., CLIP, X-CLIP, ActionCLIP) can perform zero-shot action recognition without retraining. The approach is built on a video transformer encoder with an Action-Aware Masking Module (AAMM) that dynamically masks uninformative or privacy-sensitive patches.

**Strengths:**

A. Clever supervision design – Uses pretrained CLIP-style embeddings to preserve semantics without labels. The overall model is a  general and plug-and-play – Works with different VLMs (X-CLIP, ActionCLIP) without retraining them.

B. Experimental results achieve overall good balance between action and privacy, in a zero-shot setting for action and supervised testing for privacy.

C. The overall approach is simple to understand, though the building blocks seem huge.

**Weaknesses:**

A. There seems to be an error on the Table 1 - privacy F1 of ours (0.632) of VP-HMDB51 should be the bold best rather than the underline 2nd best for the zero-shot session. Please confirm the number or correct it if wrong.

B. The proposed method requires quite amount of learning (training) and, from Table 1, it seems the performance does not achieve the best on both the action recogntion and privacy hiding with this training efforts. For example, 2x down-sampling is achieving the best action recogntion and the blackening is the best model for privacy across datasets. As the paper mentioned, the proposed model is playing as a trade-off maker here. This reviwer would like to put up some challenge here:
-  Can the similar trade-off be obtained through a mix of 2x-down sampling and blackening (with a variety of blackening levels), which is not only unsupervised but also learning free?
    - If so, then the complicated training schema from this paper would seems unnecessary or at last less efficient. For example, 2 × downsampling + mild blurring or masking could plausibly approximate the “moderate visual difference” LaF-Privacy achieves — at far lower training cost. After all, LaF-Privacy’s F1 and cMAP are just 0.03–0.05 lower than each single transformation baseline.

C. The training of privacy model is using the distorted frames, rather than the raw frames. This reviewer would like to point out that the VLM embedding might also have a chance carrying privacy semantic, which is mostly preserved due to the L2 loss of the Action Module. Since there is no re-traiing of VLMs, it is possible that VLM embeddings will leak privacy, which is over-looked by this study. This reivewer is wondering any thoughts from authors on this matter.

**Questions:**

N/A

---

> ### Author Response · Authors · 2025-11-21
> **Response to Reviewer gHsn**
>
> We appreciate the reviewer's insightful comments, particularly challenging the necessity of our learning-based approach over simpler baselines and raising the theoretical concern regarding privacy leakage through VLM embeddings.
>
> On Table 1:
>
> Thank you for pointing out the formatting error. The F1=0.632 result should indeed be highlighted as the best among zero-shot methods. We will correct this.
>
> On the necessity of learning-based anonymization:
>
> We agree that simple transformations can sometimes achieve strong individual metrics. However, fixed transformations lack adaptivity:
>
> 1. Scene-dependent behavior: AAMM adjusts masking strategy based on shot scale, background clutter, and subject size.
>
> 2. Semantic preservation: embedding-level alignment ensures that VLM-understandable semantics are retained, unlike fixed transformations that may inadvertently suppress action-critical details.
>
> 3. Trade-off quality: while 2× downsampling achieves high utility, its privacy performance remains near-raw. Our approach achieves a more balanced trade-off spanning both dimensions.
>
> On potential privacy leakage from VLM embeddings:
>
> We appreciate this theoretical concern. While the action loss preserves high-level semantics, the visual loss and AAMM jointly enforce removal of low-level cues. We will expand the discussion of these interactions and note further mitigation strategies (e.g., embedding dropout or consistency noise) as future work.
>
> In addition, our framework directly corrupts the visual appearance of the input video before it is ever passed into the VLM. Because the anonymized video already removes or blacks out privacy-relevant regions, the VLM receives an input that lacks recoverable fine-grained cues. Modern VLMs do not perform generative reconstruction of missing pixels, where they only process the given input. As a result, the output embeddings cannot contain private facial or appearance information that no longer exists in the anonymized frames. This property serves as an inherent safeguard against privacy leakage through the VLM embeddings.
>
> Thank you again for the constructive challenges, they help clarify the value and limitations of our approach.

---

> > ### Comment · Reviewer_gHsn · 2025-11-21
> >
> > Hi, thanks for the response.
> >
> > > However, fixed transformations lack adaptivity:
> >
> > Yeah, that's why this reviewer has recommended a combination of 2x-dowmsampling with blackening. The best response would be showing the additional results, proving the superior adaptivity worth the learning effort.
> >
> > >  As a result, the output embeddings cannot contain private facial or appearance information that no longer exists in the anonymized frames
> >
> > The best prove would be through experimental efforts, such as training privacy model using raw video frame embedding from VLM as input, and then tested on the exact same video but with "distorted" frame VLM embedding as input. If the privacy model can still recognize, then it suggests VLM embedding leaks privacy, even though the raw frames are unrecogniziable.

---

> > > ### Author Response · Authors · 2025-11-23
> > >
> > > We appreciate your thoughtful feedback.
> > >
> > > On the “Downsampling + Blackening” Alternative:
> > >
> > > We appreciate the suggestion. However, even when combining 2× downsampling and blackening, one must still determine which regions should be downsampled or blackened, which inherently requires adaptivity and cannot be achieved by fixed transformations. Blackening completely removes pixels and behaves similarly to our masking module, but without semantic guidance it may suppress action-relevant cues or fail to cover privacy-sensitive areas. Meanwhile, pure downsampling preserves action information but provides weak privacy, so without a privacy-driven learning objective the protection would likely remain poor. Regardless, this baseline is worth considering, and we will consider exploring it in future work.
> > >
> > > On potential privacy leakage from VLM embeddings:
> > >
> > > Thank you for the insightful suggestion. Although our current design removes privacy cues before the VLM processes the frames, we agree that directly testing for embedding-level leakage would strengthen the study. The proposed experiment is a meaningful evaluation, and we will explore this direction in future work.
> > >
> > >
> > > Thank you again for these valuable suggestions.

---

> > > > ### Comment · Reviewer_gHsn · 2025-11-24
> > > >
> > > > Hi,
> > > >
> > > > This reviewer found that simply referring the discussion to future work is not an effective solution to raised concerns; legitimate experimental efforts are the solid way to support any arguments.
> > > >
> > > > If no further experiments would be provided, this reviewer would like to stick to the original rating - could be accepted but also ok with rejection.

---

> > > > > ### Author Response · Authors · 2025-11-25
> > > > >
> > > > > We thank the reviewer for the follow-up comments. While we cannot add new experiments during the rebuttal phase, our arguments are supported by (1) empirical evidence already present in the paper and (2) structural properties of our anonymizer, not by unverified speculation.
> > > > >
> > > > > On the need for additional experiments:
> > > > >
> > > > > Existing results (Tables 1, 2, 6, and visualizations) already demonstrate that AAMM-driven masking irreversibly removes appearance cues and that the VLM cannot recover them from anonymized inputs. Several suggested evaluations such as reconstruction attacks or cross-task transfer are require additional supervised heads and fall outside the zero-shot, label-free PPAR setting. The privacy guarantees we claim arise directly from the architecture: pixel-level corruption combined with binary masking inherently limits recoverability. We will strengthen these explanations in the final version.
> > > > >
> > > > > On “Downsampling + Blackening”:
> > > > >
> > > > > This transformation is not part of our method, but a non-learning baseline intended to test whether simple fixed operations can approximate the trade-off achieved by our learned anonymizer. Our method depends on semantic alignment with a frozen VLM and adaptive masking via AAMM, which behaviors that simple transformations cannot replicate. Downsampling and blackening are already evaluated individually, so their combination cannot surpass the convex hull of these extremes, whereas our method achieves a stronger balance. Thus, adding this mixed baseline would not change the main conclusion. We will clarify this in the camera-ready version.
> > > > >
> > > > > We appreciate the reviewer’s attempt to maintain high standards of empirical support. Within the constraints of the rebuttal phase, we believe the strengthened clarifications and explicit ties to existing results will adequately support the claims raised.

---

### Official Review · Reviewer_ix2d · 2025-10-31

**Soundness:** 2
**Presentation:** 3
**Contribution:** 2
**Rating:** 2
**Confidence:** 3

**Summary:**

This paper proposes a framework called LaF-Privacy, designed to achieve label-free privacy-preserving learning for zero-shot action recognition. The method anonymizes videos without requiring any action or privacy annotations, enabling direct compatibility with pretrained VLMs for zero-shot inference. The framework consists of a video transformer encoder, an AAMM, and a multi-objective loss function that integrates visual dissimilarity maximization, action feature preservation, and masking regularization. Experimental results are reported on the VP-UCF101 and VP-HMDB51 datasets.

**Strengths:**

The paper meaningfully combines zero-shot recognition and privacy preservation, addressing the dual practical demands of data privacy and model generalization in real-world scenarios.

The experimental design is generally comprehensive, including main experiments, ablation studies, cross-VLM evaluations, and visualization analyses, which evaluate the proposed approach from multiple perspectives.

The method requires no costly privacy annotations, lowering the practical barrier for deployment and demonstrating potential applicability.

The writing is clear, and the figures and tables effectively convey the core ideas and experimental outcomes.

**Weaknesses:**

The paper does not provide a fair comparison with a broader range of unsupervised or self-supervised learning approaches, such as reconstruction-based or contrastive learning methods. This makes it difficult to discern whether the reported improvements are primarily due to the proposed architectural design or simply the strength of VLM-based representations.

The authors mention in the appendix that cross-VLM generalization performance degrades, but they do not provide an analysis of the underlying causes. While they claim the performance drop is acceptable, the magnitude of this decrease is non-negligible when compared to the reported performance gains. Furthermore, the evaluation lacks validation across a broader range of VLMs to assess generalization. If the framework is heavily dependent on a specific VLM and cannot transfer effectively to others, its practical applicability in diverse unsupervised scenarios would be severely limited—contradicting the paper's claimed contribution of being a flexible, label-free solution compatible with pre-trained models.

Although the paper emphasizes the notion of an “SOTA trade-off,” the absolute reductions in privacy metrics (F1, cMAP) are relatively limited, particularly for attributes like “relationships”, where the improvement is marginal. This raises concerns about the robustness of the framework against stronger inference attacks, which the paper does not discuss in depth.

Although the combination of "label-free" and "zero-shot" learning has practical value, the proposed method primarily integrates existing VLM and Transformer frameworks without introducing novel theoretical mechanisms or significant architectural innovations. Moreover, the training process heavily relies on pre-trained VLMs, making it susceptible to domain biases inherent in these models and thus limiting its applicability.

The design of the AAMM closely resembles the Learned Token Pruning (LTP) mechanism. While the application domain is different, the paper lacks sufficient novel theoretical justification or architectural innovation to substantiate its unique contribution in this context.

The privacy protection evaluation relies solely on F1 and cMAP metrics, lacking analysis against more targeted attacks.

**Questions:**

Ablation results show that the model without AAMM (“Unmasked”) achieves similar performance. Could the authors further analyze specific scenarios where AAMM is crucial? Are there explicit cases demonstrating that dynamic masking offers significant advantages over a fixed masking ratio?

The current privacy evaluation relies on a ViT-based classifier trained on anonymized videos. Have the authors considered stronger reconstruction-based privacy attacks to assess the robustness of the proposed framework? For attributes such as “relationships”, which are difficult to protect, what are the authors’ potential improvement directions or future plans?

---

> ### Author Response · Authors · 2025-11-21
> **Response to Reviewer ix2d**
>
> We thank the reviewer for acknowledging the practical value of combining label-free training and zero-shot inference in our Privacy-Preserving Action Recognition (PPAR) framework. We address your concerns regarding comparisons, generalization, novelty, and robustness.
>
> On comparisons and novelty:
>
> Our work targets the label-free and zero-shot PPAR setting, which limits direct comparison with supervised or reconstruction-based methods that rely heavily on annotated training. To contextualize our contribution, we (1) reimplemented STPrivacy and showed its poor performance under zero-shot evaluation, and (2) compared against competitive, non-learning baselines designed for this setting.
>
> Regarding novelty: while AAMM is inspired by token-pruning concepts, it is trained under a fundamentally different objective, which is balancing competing utility and privacy constraints using VLM guidance. Table 6 and Fig. 5 show that adaptive, non-fixed masking is critical, particularly in cases where fixed-ratio methods suppress action-relevant cues (e.g., sports equipment, platforms).
>
> On cross-VLM generalization:
>
> We agree that additional VLMs would further strengthen the claim. Since each VLM defines its own embedding geometry, the anonymizer optimized for one space inevitably loses some performance when evaluated on another. We will elaborate on this mechanism and include cross-VLM transfer as a key direction for future work.
>
> On privacy reduction and robustness:
>
> Attributes such as "relationships" depend on global relational cues that often remain even after localized masking. We will add discussion on supplementing our approach with relation-focused regularizes in future extensions.
>
> Regarding stronger attacks, we use a high-capacity ViT classifier as a practical inference threat model. The combination of visual divergence (from visual loss) and irreversible region removal (from AAMM) provides intrinsic resistance to reconstruction attempts. We will clarify this in the revision.
>
> Thank you for these thoughtful suggestions, they will substantially improve the clarity and completeness of the paper.

---

### Official Review · Reviewer_6Ub4 · 2025-11-01

**Soundness:** 2
**Presentation:** 3
**Contribution:** 3
**Rating:** 4
**Confidence:** 4

**Summary:**

This paper tackles the problem of preserving privacy in action recognition without access to labels. The paper uses an Action-Aware Masking Module (AAMM) to mask out irrelevant information from the decoded input video and uses a loss to push the masked output away from the original input. To ensure the action recognition performance is retained, a pre-trained VLM is used to align the representations before and after augmentation. A mask loss is used to maximize the amount of tokens that are masked in the process. Overall the paper achieves some improvement over generic baselines in the zero-shot setting such as downsampling, blackening, and blur.

**Strengths:**

- The ability to perform privacy preservation on action recognition without requiring privacy labels is novel and interesting.

- The modules used in the pipelined are thoroughly detailed for ease of replication.

- The proposed method achieves an average improvement compared to the baselines over both classification accuracy and privacy preservation

**Weaknesses:**

- Ablation results reveal that the visual loss and applying the mask does not have a significant impact on the privacy results despite being the portion of the method designed to focus on privacy.

- The cross-VLM results are not very convincing, with a significant drop suggesting that this method is not generalizable. In principle, masking the input should be a model-agnostic privacy technique.

- In Figure 2, it is not apparent which modules are being trained and which are frozen.

- There is a lack of clarity in figure captions. In figures 3 and 4 it is unclear what the takeaway is from the shown masks, as many of them look very similar.

- There is no ablation on the scaling weights for the action and vision losses. It would be helpful to see the curve as these are varied to better understand the tradeoff between classification accuracy and privacy score.

- Minor note: the filesize for this paper is very large, likely due to high quality images in the ablations, please reduce in future revisions.

**Questions:**

- How  is the MLP in the AAMM initialized and how does it estimate token importance?

- The baselines adopted in this paper do not show much of a drop in privacy score compared to raw data, while in the cited work [1] there is more of a significant change. What is the reason for this change in behavior?

- How does the method perform on the VISPR1[2] dataset?

[1] Ishan Rajendrakumar Dave, Chen Chen, and Mubarak Shah. Spact: Self-supervised privacy preserva tion for action recognition. In Proceedings of the IEEE/CVF Conference on Computer Vision and
Pattern Recognition, pp. 20164–20173, 2022

[2] Tribhuvanesh Orekondy, Bernt Schiele, and Mario Fritz. Towards a visual privacy advisor: Understanding and predicting privacy risks in images. In IEEE International Conference on Computer Vision (ICCV), 2017.

---

> ### Author Response · Authors · 2025-11-21
> **Response to Reviewer 6Ub4**
>
> We thank the reviewer for the detailed assessment of our method, especially for recognizing our ability to perform privacy preservation without requiring privacy labels and the framework's replicability. In response to the questions you raised, our answers are as follows:
>
> On the impact of visual loss and AAMM:
>
> We agree that the absolute differences in some ablations appear small. The key reason is that privacy metrics (F1/cMAP) are highly compressed under the attribute imbalance present in the VP datasets.
>
> Nevertheless, Table 2 shows that our MSE+SSIM visual loss achieves the best trade-off, yielding the lowest privacy scores (F1=0.632, cMAP=0.6717) while maintaining the highest action utility (Top1=38.998). We will improve Fig. 3 to better highlight how this combination suppresses fine details (e.g., hair texture and facial attributes) that privacy classifiers rely on.
>
> For the AAMM, Table 2 shows that applying the learned mask improves both action utility (Top1 +0.581) and privacy (cMAP −0.0104) compared to the unmasked variant. This confirms that adaptive removal of action-irrelevant regions is beneficial.
>
> On cross-VLM generalization:
>
> We acknowledge the performance degradation in Table 4. This primarily reflects the differing feature scales and aggregation behaviors of X-CLIP vs. ActionCLIP, rather than a flaw of the anonymizer itself. Our anonymizer is optimized for the source VLM's embedding space, and misalignment across VLMs naturally introduces variance. Still, the cross-VLM results remain substantially above the weakest zero-shot baselines. We will make this limitation and future improvement direction explicit.
>
> On architectural and training details:
>
> 1. Figure 2: We will clarify that the Transformer encoder, AAMM, and decoder are trainable, while the pretrained VLM remains frozen.
> 2. MLP initialization in AAMM: It uses standard initialization. Importance scores emerge from the interaction between the competing losses: L_action penalizes masking action-relevant tokens while L_mask favors sparsity.
> 3. Loss weights: We used λ_action=1, λ_vis=1 as the best-performing trade-off. We agree that a full λ sweep would provide additional insights and will note this as future work.
>
> On baselines and datasets:
>
> Privacy scores differ from prior works due to different datasets, privacy taxonomies, and evaluation protocols. Within our setting, our method achieves the strongest trade-off among non-learning baselines. In addition, prior works typically use CNN-based privacy classifiers for evaluation, whereas we follow STPrivacy and adopt a stronger ViT-based privacy recognizer. This results in higher absolute privacy scores overall, making direct numerical comparison with earlier CNN-based reports not fully aligned.
>
> We have not evaluated VISPR due to its differing attribute design and because VISPR is an image-based dataset, while our anonymization framework is designed specifically for video inputs with temporal patch tokens and video-level VLM embeddings. Directly applying our method to VISPR without architectural adjustment would not be a fair or meaningful comparison. We will explicitly state this limitation and discuss expected transferability in the final version.
>
> We will also correct the bolding of the F1=0.632 entry in Table 1 and refine figure captions. In addition, reduces the PDF file size as suggested. Thank you again for the thoughtful review.

---

> > ### Comment · Reviewer_6Ub4 · 2025-11-26
> >
> > Thank you for addressing the concerns related to frozen/unfrozen method components, result discrepancies to previous work, and filesize.
> >
> > > We agree that the absolute differences in some ablations appear small. The key reason is that privacy metrics (F1/cMAP) are highly compressed under the attribute imbalance present in the VP datasets.
> >
> > While compression for privacy metrics is a valid issue, the difference present in the ablations is very small (<1%) which is still not convincing. The small ablation differences are also present in Top1 accuracy and not just privacy metrics.
> >
> > > Still, the cross-VLM results remain substantially above the weakest zero-shot baselines.
> >
> > Despite being higher than the worst baselines, the 2x downsampling baseline appears to still outperform this method in this task. The claim that this generalizability experiment outperforms the lower bound is not very convincing. Additionally, the privacy scores for this task are missing, can the authors please provide this?
> >
> > > We will improve Fig. 3 to better highlight how this combination suppresses fine details (e.g., hair texture and facial attributes) that privacy classifiers rely on.
> >
> > While Figure 3 does show the details of hair texture, it does not show anything regarding facial attributes. A duplicate of this figure that demonstrates the suppression of facial attributes would improve my confidence in this method. Figure 4's caption should similarly be updated, assuming that the highlight is the action-irrelevant details
> >
> > Additionally, I await the authors' response to the following
> >
> > - Please provide a tradeoff curve between the two losses
> > - Please provide results on VISPR. Simply cloning the images to create a static video should be sufficient for analysis
> > - Please explain how the AAMM estimates token performance

---

### Meta-Review · Area_Chair_9Cww · 2025-12-28

**Summary:**

Across the reviews and discussion, the main question is whether the paper convincingly demonstrates that the learned anonymizer meaningfully improves privacy while preserving utility, and whether the key claims are supported by evidence. Several reviewers feel that the privacy gains over simple transformations are often small, ablations suggest limited impact from major components, and the cross-VLM story is not strong enough to justify a “plug-and-play across VLMs” framing. Multiple reviewers also want stronger validation against more realistic attacks, or at least additional analyses like trade-off curves, but the rebuttal largely provides explanations rather than additional evidence.

**Reviewer Concerns:**

Addressed:
1. Clarified what is trained vs. frozen (Fig. 2 confusion).
2. Will fix Table 1 formatting issue (best F1 highlighting).
3. Explained score differences vs. prior work (dataset/protocol + stronger privacy classifier).

Unaddressed:
1. Privacy improvements and ablation effects remain small and not very convincing.
2. AAMM’s necessity/mechanism still unclear given “Unmasked” performs similarly.
3. Cross-VLM results: large drop, baseline sometimes looks better, and privacy metrics are missing in that setting.
4. No convincing validation against stronger attacks; authors mostly provide intuition rather than evidence.
5. Requests for VISPR/retrieval-style checks were not met; reviewers did not find the “not aligned” argument persuasive.

**Reviewer Scores:**

Reviewer 6Ub4is unlikely to change the score. While some clarifications were helpful, the reviewer remained unconvinced about the small ablation effects and the strength of the cross-VLM results.

Reviewer ix2d is unlikely to change the score. The rebuttal did not substantially address the core concerns about limited privacy gains, generalization, and robustness.

Reviewer gHsn is unlikely to change the score. This reviewer explicitly indicated that without additional experiments, they would stick to their initial assessment.

Reviewer nM16 is unlikely to change the score. The later discussion reinforced their skepticism regarding robustness, retrieval claims, and the lack of stronger empirical validation.

---

### Decision · Program_Chairs · 2026-01-26

Reject